# Nutrition, Lifestyle, and Environmental Factors in Lung Homeostasis and Respiratory Health

**DOI:** 10.3390/nu17060954

**Published:** 2025-03-09

**Authors:** Athanasios Pouptsis, Rosa Zaragozá, Elena R. García-Trevijano, Juan R. Viña, Elena Ortiz-Zapater

**Affiliations:** 1Department of Biochemistry and Molecular Biology-INCLIVA, University of Valencia, 46010 Valencia, Spain; atpoupt@alumni.uv.es (A.P.); elena.ruiz@uv.es (E.R.G.-T.); juan.r.vina@uv.es (J.R.V.); 2Department of Human Anatomy and Embryology-INCLIVA, University of Valencia, 46010 Valencia, Spain; rosa.zaragoza@uv.es

**Keywords:** lungs, homeostasis, respiratory function, metabolism, micronutrients, diet, chrono nutrition, lifestyle, pollutants

## Abstract

The lungs play a vital role in maintaining homeostasis by facilitating gas exchange and serving as a structural and immune barrier. External factors, including nutrition, lifestyle, and environmental exposures, profoundly influence normal lung function and contribute to the development, progression, and prognosis of various respiratory diseases. Deficiencies in key micronutrients, such as vitamins A, D, and C, as well as omega-3 fatty acids, can impair the integrity of the epithelial lining, compromising the lungs’ defense mechanisms and increasing susceptibility to injury and disease. Obesity and physical inactivity further disrupt respiratory function by inducing structural changes in the chest wall and promoting a pro-inflammatory state. Environmental pollutants further worsen oxidative damage and activate inflammatory pathways. Addressing these modifiable factors through interventions such as dietary optimization, physical activity programs, and strategies to reduce environmental exposure offers promising avenues for preserving lung function and preventing disease progression. This review examines the molecular pathways through which nutrition, lifestyle, and environmental influences impact lung homeostasis.

## 1. Introduction

The lungs are vital organs, primarily responsible for gas exchange, essential for maintaining oxygen delivery to tissues and removing carbon dioxide from the body. This central role supports cellular metabolism and helps regulate the acid-base balance, ensuring the physiological stability necessary for overall health. Nevertheless, the lungs perform functions beyond respiration, serving as structural and immunological barriers and protecting against pathogens and pollutants. Despite their importance, the metabolic functions of the lungs are often overlooked, even though research has shown that their glucose consumption surpasses that of organs like the heart or the liver. Most lung cells dedicate their energy to essential cellular processes, such as mRNA transcription and protein synthesis. However, certain specialized cells in the lung, such as those responsible for the rhythmic beating of cilia or the production of surfactants, are engaged in energy-intensive activities crucial for maintaining pulmonary function.

External influences, including diet, lifestyle choices, and environmental exposures, play a significant role in maintaining or disrupting lung health. Insufficient intake of essential micronutrients—such as vitamins A, D, and C, along with omega-3 fatty acids—weakens the protective epithelial barrier, diminishing the lungs’ defense against pathogens and environmental insults while increasing vulnerability to damage and disease. At the same time, conditions like obesity and physical inactivity contribute to respiratory dysfunction by altering chest wall mechanics and fostering a state of chronic inflammation. Environmental contaminants, including particulate matter, ozone, and nitrogen oxides, amplify oxidative stress and inflammatory responses, further accelerating lung tissue damage and the development of respiratory conditions. Although the impact of these external factors on lung function is well established, the precise molecular and cellular mechanisms underlying their effects remain insufficiently explored.

In this review, we aim to explore the metabolic pathways in the lung and examine how cells adapt to energy deficits. We will begin with a brief overview of the lung’s structure and anatomy to contextualize the discussion on cellular energy consumption and metabolic pathways. Additionally, we will analyze how dietary habits, lifestyle choices, and environmental factors affect lung metabolism, emphasizing the potential role of nutritional supplements in preventing or reversing metabolic dysfunction. Interestingly, while metabolic dysfunction has been widely recognized as a contributing factor to many extrapulmonary diseases, its role in respiratory diseases has only recently gained attention. Addressing these modifiable factors through interventions such as dietary optimization, physical activity, and reducing exposure to environmental pollutants could offer promising strategies to preserve lung function and prevent disease progression.

Finally, we will review emerging evidence that links alterations in cellular metabolism to the pathobiology of common respiratory diseases, including chronic obstructive pulmonary disease (COPD), asthma, and pulmonary fibrosis. By understanding the molecular pathways through which nutrition, lifestyle, and environmental factors impact lung health, we can identify novel strategies for prevention and treatment, improving outcomes for patients with respiratory diseases.

## 2. The Lung: Anatomy, Cell Types, and Metabolic Flexibility

The respiratory tract is a highly intricate organ system, anatomically divided into the upper and lower airways. The upper portion includes the nose, nasal passages, pharynx, and larynx, while the lower respiratory tract comprises the trachea, bronchi, and bronchioles, which form the conducting airways essential for respiration. It extends to the respiratory bronchioles, alveolar ducts, alveolar sacs, and alveoli, where gas exchange occurs. The respiratory system is composed of a diverse range of cell types, including epithelial cells and resident lung cells such as immune and mesenchymal cells, all interconnected by the extracellular matrix (ECM) [1]. The airway epithelium consists of a diverse community of cells, each with unique molecular features. Traditionally, four main cell types have been identified within the airway epithelial layer: secretory club cells, goblet cells, ciliated cells, and basal cells, that line the conduction airways. Each cell type has specific functions, determining its requirements and defining its main metabolic pathways (See Figure 1). With the advent of single-cell RNA sequencing (scRNA-seq), additional subsets of airway epithelial cells have been discovered, revealing that the cellular composition of the airway epithelium is far more complex and diverse than previously recognized. These findings underscore the variability in cell populations depending on the specific region of the respiratory tract, reflecting the structural and functional differences across the lungs [2]. This diversity also defines the main catabolic and anabolic pathways used by each cell. In that sense, other cells such as Hillock cells (with barrier and immunomodulation functions), Tuft cells (with stem cells properties), ionocytes, or pulmonary neuroendocrine cells (PNECs) have been identified and extensively described [3,4,5]. In the lower airway, where gas exchange occurs, the epithelium is formed by types I and II pneumocytes, with alveolar macrophages serving as the main resident immune cell. Type I pneumocytes are thin, flat cells that form most of the alveolar epithelium facilitating gas exchange. They rely on oxidative phosphorylation for energy to sustain their thin structure and support efficient gas exchange. Type II pneumocytes are cuboidal cells interspersed among type I pneumocytes. These cells play a crucial role in surfactant production and alveolar repair, and rely on glycolysis and oxidative phosphorylation for energy, with enhanced lipid metabolism to synthesize surfactant and support cellular repair.

We have briefly summarized its main functions and the specific metabolic patterns of each epithelial cell type (see Figure 1). It is also crucial to consider the roles of resident immune cells, such as macrophages, and structural cells like fibroblasts. Macrophage metabolism has been extensively studied, revealing that these cells can undergo metabolic reprogramming to fulfill various functions. For instance, pro-inflammatory (M1) macrophages primarily rely on glycolysis, while anti-inflammatory (M2) macrophages depend more on oxidative phosphorylation [6]. This metabolic flexibility allows macrophages to adapt to different microenvironments and perform a range of activities, from pathogen clearance to tissue repair. Similarly, fibroblasts exhibit distinct metabolic profiles that influence their activation and function. Upon activation, fibroblasts increase their glycolytic function to support rapid proliferation and extracellular matrix production. Notably, metabolic reprogramming in fibroblasts has been implicated in the development of fibrosis [7].

## 3. Main Metabolic Pathways in the Lung

Commonly recognized for facilitating gas exchange, the lungs are also highly metabolically active organs, with energy requirements that match or surpass those of critical organs like the brain and heart. Lung metabolism powers not only routine cellular functions, like protein synthesis and DNA repair, but also specialized processes critical for respiratory health, such as surfactant production, ciliary motion, and airway clearance. The lungs rely heavily on glycolysis, fatty acid oxidation, and other metabolic pathways to meet these diverse energy requirements (Figure 2). Recent research advances have uncovered how these metabolic processes adapt to physiological changes and contribute to both lung health and disease [8].

### 3.1. Catabolic Pathways in the Lungs

Glycolysis and the Warburg Effect: In most tissues, including the lungs, glucose is the main source of energy [9]. Its metabolism yields two pyruvate, two ATP, and two NADH molecules, supporting various biological processes. Once pyruvate is generated, it is either metabolized in mitochondria via the tricarboxylic acid cycle (TCA) or converted to lactate in the cytoplasm by the enzyme lactate dehydrogenase (LDH). Notably, a significant portion of glucose is converted to lactate even under normal oxygen conditions, suggesting that aerobic lysis, similarly to cancer cells and the Warburg effect [10,11], may have evolved to optimize oxygen availability for other tissues to be used [12]. Lactate may also serve as an energy source for lung cells with limited nutrient access, mirroring tumor microenvironment mechanisms [13].

The pentose phosphate pathway (PPP): The PPP, parallel to glycolysis, generates NADPH, essential for cellular processes such as glutathione regeneration and lipid synthesis. While NADPH can also be produced via malic enzyme or isocitrate dehydrogenase, the lungs primarily rely on the PPP [14]. Additionally, the PPP is responsible for the isomeric conversion of ribose 5-phosphate, necessary for nucleotide synthesis making the PPP vital for all active and dividing cells [12,15].

The Tricarboxylic Acid Cycle (TCA) cycle maximizes the ATP production primarily using acetyl-CoA from glucose or fatty acids. Amino acids (via pyruvate) also contribute as substrates [16]. Beyond energy production, the TCA cycle supports biosynthesis through cataplerosis. For instance, citrate can be exported for fatty acid and phospholipid synthesis, essential for pulmonary surfactant production [17].

Fatty acid β-oxidation: To maintain glucose levels, fatty acids can be released from lipid droplets via lipolysis or lipophagy and utilized for energy production through β-oxidation and the TCA cycle in the mitochondria. While the lung primarily relies on glucose metabolism, β-oxidation increases under fasting, metabolic stress, or nutrient deprivation. Shaw and Rhoades reported a 40% rise in fatty acid β-oxidation in rat lungs under starvation [18]. Additionally, alveolar type II cells utilize fatty acids for surfactant synthesis rather than energy.

Glutamine metabolism: Finally, glutamine can serve as an alternative metabolic fuel, helping cells—especially rapidly proliferating ones—meet their increased demand for ATP. It is converted into glutamate and subsequently into α-ketoglutarate, which feeds into the TCA cycle, supporting efficient ATP generation [17]. Recently, it has been seen that glutamine metabolism is required for alveolar epithelial regeneration during lung injury [18].

### 3.2. Lipid Synthesis in the Lung

In the lungs, lipid synthesis has often been considered specific to surfactant-producing cells such as Alveolar type II pneumocytes and also Club cells in mouse lungs [19]. Lipid synthesis is an energy-intensive process, and it is tightly linked to the cell’s energy status. For example, elevated ATP levels promote lipid synthesis through the upregulation of enzymes like fatty acid synthase and acetyl-CoA carboxylase (ACC) and transcription factors such as sterol regulatory element-binding protein and liver X receptor (LXR). Biosynthetic intermediates like pyruvate (from glycolysis) and citrate (from the TCA cycle) are also used.

In type II alveolar epithelial cells, de novo lipid synthesis utilizes pyruvate or citrate mobilized from mitochondria. These cells must sustain lipid production even under metabolically unfavorable conditions like starvation, due to high turnover of surfactant phospholipids in the lung [20]. When fatty acid synthesis is limited by high AMP/ATP ratios, these cells rely on alternative methods to replenish surfactant lipids, including recycling old surfactant lipids through intracellular remodeling and resecretion and incorporating circulating fatty acids into intracellular glycerol backbones [21]. These mechanisms allow type II cells to maintain surfactant lipid levels while supporting other energy-intensive activities, such as producing surfactant proteins, detoxifying chemicals, and regenerating epithelial cells.

## 4. How Nutrition Affects Lung Metabolism

Metabolic pathways are highly regulated and rely heavily on the availability of substrates present in the organism. Since the main goal of metabolism is to maintain glucose levels in the bloodstream, it is evident that metabolism in general—and respiratory metabolism specifically—are significantly influenced by dietary intake and the availability of both macronutrients and micronutrients. When metabolic homeostasis is disrupted, for example in malnutrition or aggressive pharmacological treatments, lung function can be adversely affected by diminished respiratory muscle strength, altered ventilatory capacity, and impaired immune function. But to what extent does our diet affect the lung function in healthy individuals? A recent study examined the association between macronutrient intake and lung function in healthy adults (*n* = 5880) using the Ansan–Ansung cohort study [22]. The findings showed that lung function improved by high protein and fat intake but worsened by high carbohydrate intake, with age and obesity playing key roles. In terms of micronutrients, the data are inconclusive. A study looking at the associations between vitamin A and K intake and lung function in the general US population concluded that greater vitamin A or K intake was independently associated with better lung function assessed by spirometry [23]. However, micronutrient supplementation, such as vitamin D alone and compound nutrients, has been shown to improve lung function in patients with COPD [24]. In the following sections we will explore in detail the potential role of nutritional supplements in both healthy individuals and respiratory patients.

### 4.1. Micronutrients and Metabolic Regulation

The role of vitamins has been recognized for over two centuries, primarily due to the detrimental effects associated with their deficiency. Micronutrients, including vitamins and minerals, have various biochemical functions, such as cofactors, coenzymes, antioxidants, and regulators of gene expression. In the lungs, they have an established role in maintaining epithelial barrier integrity, modulating the immune response, and mitigating oxidative stress [25]. In this section, we describe the potential health benefits of vitamins and omega-3, focusing on their use in contexts that extend beyond merely addressing deficiencies.

Vitamin A: The importance of Vitamin A in maintaining the integrity of the respiratory epithelium is well-established [26,27]. Its deficiency impacts normal lung morphology by modifying the extracellular matrix and basement membrane, reducing elasticity and structural integrity [28]. Vitamin A is essential for normal ciliary function, the regulation of mucus production, and to maintain the structural elasticity and the tensile strength required for lung function. It also regulates various immune processes, such as lymphocyte activation, T-helper cell differentiation, and the production of specific antibody isotypes. The biological effects of Vitamin A are primarily mediated through its active metabolite, all-trans-retinoic acid (ATRA), a potent regulator of gene expression. ATRA orchestrates the formation and repair of the respiratory epithelium and basement membrane through different pathways, such as the TGF-β and Wnt signaling cascades, which are integral to lung morphogenesis during embryogenesis and support tissue homeostasis in adulthood. ATRA also triggers kinase cascades, including the p38, MAPK, and ERK pathways, that are crucial for alveolar epithelial repair, immune cell signaling, and maintaining a balanced response to oxidative stress [29,30]. When talking about vitamin supplements, it is more relevant to describe the CARET study (Beta-Carotene and Retinol Efficacy Trial), that examined whether beta-carotene and vitamin A supplements could prevent lung cancer and cardiovascular disease in high-risk groups, including smokers and asbestos-exposed individuals [31]. The trial found that supplementation not only failed to provide protection but was associated with an increased risk of lung cancer and mortality. Its results highlighted the potential dangers of high-dose antioxidant supplementation in certain populations and the importance of maintaining correct levels. Since then, not many studies have explored the administration of vitamin A in lung diseases. Checkely and collaborators examined a cohort of rural Nepali children 9 to 13 years of age whose mothers had participated in a placebo-controlled, double-blind, cluster-randomized trial of vitamin A or beta-carotene supplementation between 1994 and 1997. They concluded that in a chronically undernourished population, maternal repletion with vitamin A at recommended dietary levels before, during, and after pregnancy improved lung function in offspring [32]. It has also been shown that vitamin A supplementation prevents the bronchopulmonary dysplasia in premature infants [33]. Finally, a more recent study showed that a higher intake of preformed vitamin A, but not β-carotene, in mid-childhood is associated with higher subsequent lung function and lower risk of fixed airflow limitation and incident asthma [34].

Vitamin D: Vitamin D is another fat-soluble micronutrient that is essential for calcium homeostasis and bone health, but also for the lungs [35]. Once synthesized or ingested, vitamin D is converted to its active form, calcitriol, the high-affinity ligand for the vitamin D receptor (VDR) In the lungs, vitamin D plays a crucial role in maintaining immune homeostasis, as it has a protective role against respiratory infections, and modulates inflammatory responses [36]. Several studies have demonstrated that adequate levels of vitamin D reduce the risk of respiratory diseases, such as asthma, COPD, and pneumonia [24,35,37]. The active form of vitamin D, calcitriol, regulates the production of antimicrobial peptides in the lung, such as cathelicidin, which plays a key role in the defense against respiratory pathogens [38]. Vitamin D also influences pulmonary epithelial cell function by regulating cellular proliferation epithelial-to-mesenchymal transition (EMT) processes, and apoptosis [39].

Vitamin C: Vitamin C, or ascorbic acid, is a water-soluble essential micronutrient that must be obtained through the diet, primarily from fruits and vegetables, as humans are unable to synthesize it endogenously. In the lungs, vitamin C plays a critical role in protecting against oxidative stress, which is a key factor in the pathogenesis of various pulmonary diseases such as COPD, asthma, or acute respiratory infections [37]. As the lung is particularly vulnerable to oxidative stress due to constant exposure to environmental pollutants and infectious agents, the role of Vitamin C in protecting the lung tissue is especially relevant [40]. Beyond its antioxidant activity, vitamin C is essential for maintaining the structural architecture of the respiratory system, particularly through its role in collagen synthesis, a major component of the ECM [41]. Additionally, vitamin C also modulates immune responses in the lung, especially in response to infections [42]. Recent studies have shown that vitamin C supplementation can reduce the incidence and severity of respiratory infections, particularly in individuals exposed to environmental stress or those with vitamin C deficiency [36,43]. The preventive effect of vitamin C was also demonstrated in different respiratory diseases, including asthma, COPD, lung fibrosis or cancer [44,45,46]. However, while supplementation can be beneficial in certain contexts, more research is needed to better define the optimal dosages and to understand the long-term effects of vitamin C on pulmonary health and disease prevention [47].

Omega-3-fatty acids: Omega-3 fatty acids, primarily eicosapentaenoic acid (EPA) and docosahexaenoic acid (DHA), are polyunsaturated fatty acids with anti-inflammatory and antioxidant properties. The simplest omega-3 fatty acid is α-linolenic acid, that cannot be synthesized by the human body and must be obtained through diet. Omega-3 fatty acids play a crucial role in the regulation of lung inflammation, oxidative stress, and immune function [48]. In the lungs, they modulate eicosanoid production, particularly resolvins and protectins, which help resolve inflammation and reduce tissue damage in respiratory diseases such as asthma, COPD, fibrosis, and acute respiratory infections [49,50,51]. Omega-3 fatty acids help maintain pulmonary health by regulating inflammatory responses to allergens and pollutants. Studies show that diets rich in omega-3s decrease pro-inflammatory cytokines, chemokines, and adhesion molecules in the lungs, reducing asthma severity. [49]. Additionally, omega-3 fatty acids play a role in protecting lung tissue from oxidative damage [52], cancer growth [53], and ameliorating pulmonary function [54], making them a valuable component of dietary strategies aimed at improving pulmonary health. It is important to note that the balance between omega-3 and omega-6 fatty acids is crucial for maintaining optimal health. Historically, human diets had an omega-6 to omega-3 ratio of approximately 1:1. However, modern Western diets often exhibit ratios between 10:1 and 30:1, favoring omega-6 fatty acids [55]. This imbalance is associated with increased inflammation and a higher risk of chronic diseases [56]. While the optimal ratio is still under investigation, a range between 1:1 and 4:1 (omega-6 to omega-3) is often suggested for better health outcomes. To mitigate these risks, it is recommended to increase omega-3 intake.

### 4.2. Practical Dietary Recommendations for Lung Health

Maintaining healthy lungs is essential for overall well-being, and diet plays a crucial role in supporting respiratory function. Certain nutrients can reduce inflammation, enhance lung capacity, and improve overall breathing efficiency. Fruits and vegetables are rich in antioxidants, vitamins, and minerals that help combat oxidative stress and inflammation in the lungs. Berries, apples, tomatoes, and leafy greens contain flavonoids and vitamin C, which support lung function and protect against respiratory diseases. High-fiber foods like legumes, nuts, and whole grains also contribute to better lung function and reduced inflammation [57]. As previously mentioned, omega-3 fatty acids, found in fatty fish, flaxseeds, and walnuts, have anti-inflammatory properties that can benefit lung health [58]. On the other hand, trans fatty acids and excessive omega-6 fatty acids, commonly found in processed foods, should be minimized as they may enhance inflammation and worsen lung conditions [59]. Magnesium also plays a role in lung function by helping relax bronchial muscles. Foods like nuts, seeds, spinach, and avocados are excellent sources of magnesium and can support better breathing [60]. Previous studies have explored the impact of magnesium supplementation on COPD patients, with findings suggesting several potential benefits. Research by do Amaral et al. indicates that magnesium may help reduce lung hyperinflation and enhance the strength of respiratory muscles [61]. Furthermore, patients with COPD frequently experience an imbalance between energy expenditure and intake, leading to undetected undernourishment that can negatively impact respiratory function. Ensuring adequate caloric and protein intake is crucial, as malnutrition is associated with reduced fat-free mass, impaired respiratory muscle strength, and increased morbidity [62]. Proper hydration is vital for lung health as it helps thin mucus secretions, making it easier to clear the airways. In that sense, moderation of alcohol and caffeine consumption is important to avoid dehydration, which may thicken mucus and make it harder for the lungs to function efficiently. In summary, a well-balanced diet that includes antioxidant-rich foods, unsaturated fats, essential fatty acids, whole grains, and adequate hydration can significantly contribute to lung health. For patients with respiratory conditions, consulting a healthcare professional or nutritionist for personalized dietary advice is highly recommended.

### 4.3. Chrononutrition and the Impact of Meal Timing on Metabolism

When it comes to nutrition, the primary focus is often *what* we eat—calories, macronutrients, and food quality. However, *when* we eat is equally important in regulating metabolism and health, including lung health. This concept, known as chrononutrition, explores how meal timing, frequency, and fasting periods influence our body’s biological rhythms and organ function [63]. Our metabolism follows a circadian rhythm, the internal clock that regulates digestion, hormone secretion, and energy use. Disrupting this rhythm—such as by eating late at night—has been associated with metabolic disorders like insulin resistance, obesity, and inflammation [64,65,66]. Studies suggest that consuming meals earlier in the day optimizes energy efficiency and reduces fat storage, which can support respiratory health [67]. Meal frequency also plays a role. While traditional dietary advice suggests multiple small meals per day, research indicates that less frequent meals with longer fasting periods—such as in intermittent fasting (IF)—may enhance fat metabolism, improve cellular repair processes, and support longevity. Fasting windows of 12 to 16 h can help regulate blood sugar, improve gut health, and reduce the risk of chronic diseases [68]. The effect on IF specifically on lung health has been analyzed in healthy individuals. A Saudi study examined the impact of Islamic fasting during Ramadan on lung volumes and capacities in healthy individuals, showing that fasting might increase lung volumes and improve pulmonary function, potentially due to weight changes during the fasting period [69]. Another recent study investigated the effects of time-restricted eating on pulmonary function among overweight or obese women, showing that safe dietary intervention leads to significant improvements in postural balance and pulmonary function [70]. Additionally, research on IF has shown potential benefits for cardiovascular health, which may indirectly influence lung function [71]. There are not many studies that show the consequences of IF in individuals with respiratory conditions. A pilot study investigating the impact of prolonged fasting on immune mechanisms in individuals with mild asthma revealed that fasting reduced asthma-related inflammation [72]. In the case of COPD, a 2018 pilot study found that Ramadan Intermittent Fasting (RIF) had no significant impact on spirometric data in male patients with COPD [73]. On the other hand, it is well established that IF promotes autophagy, that can help clear damaged lung cells and reduce lung tissue decline in diseases like COPD and idiopathic pulmonary fibrosis (IPF) [74]. However, in patients with severe dyspnoea or hypoxemia, prolonged fasting leading to larger, less frequent meals could exacerbate postprandial dyspnea and respiratory distress. Therefore, individualized nutritional strategies should be considered, with an emphasis on balancing metabolic benefits with respiratory function.

## 5. Other Lifestyle Factors Affecting Lung Metabolism

### 5.1. Physical Activity

It is well known that physical activity plays a crucial role in maintaining health and prevents metabolic dysfunctions. In cardiovascular disease, obesity and physical inactivity rank among the most well-established risk factors, but their role in maintaining lung health remains less clear. One of the main obstacles to conducting large epidemiological studies addressing this association is the need for long-term follow-up and the presence of multiple confounding factors. In contrast, the impact of obesity and physical activity in diagnosed respiratory diseases is easier to study due to the availability of measurable clinical endpoints. Physical activity enhances lung metabolism by boosting oxygen uptake and energy production. A recent UK cohort study showed that decreased physical activity was associated with accelerated decline in Forced Expiratory Volume in one second (FEV1), Forced Vital Capacity (FVC), and FEV1/FVC ratio [75]. Similarly, in another study, healthy individuals with higher or normal cardiovascular fitness had a lower risk of developing COPD [76]. However, caution is warranted when interpreting the results as this study assessed the cardiovascular fitness instead of physical activity, suggesting potential protective effects in at-risk populations [77].

Exercise also plays a role in regulating lung-specific metabolic pathways, particularly in conditions where oxidative stress contributes to disease progression. Recent findings suggest that physical activity enhances antioxidant buffering capacity, which may help mitigate oxidative damage in chronic lung diseases. A study on IPF found that individualized exercise training improved redox balance, increased systemic antioxidant capacity, and reduced oxidative stress markers [78]. Furthermore, aerobic exercise appears to influence mitochondrial function and inflammatory responses in lung tissue. Physical activity in mice exposed to particulate matter (PM) paradoxically had a protective effect, reducing oxidative stress, apoptosis, and inflammation in lung tissue, despite increased inhalation of pollutants due to exercise-induced hyperventilation [79]. These findings suggest that exercise not only improves lung metabolism but may also confer resilience against environmental stressors and that structured exercise programs may support lung health by modulating metabolic pathways. 

Finally, the role of physical activity in improving outcomes of patients with respiratory disease is well documented. In asthma patients, higher physical activity is associated with better disease control, improved lung function, and reduced healthcare use [80]. In a longitudinal study, lighter physical activity was associated with reduced asthma symptoms in middle-aged adults over time, while higher-intensity exercise did not show the same benefit. Additionally, results varied depending on BMI, suggesting a complex relationship between physical activity, body weight, and asthma severity [81]. The benefits of exercise and pulmonary rehabilitation are well established in patients with COPD, with strong evidence supporting their role in improving survival and quality of life. Physical activity is recommended as it has been shown to reduce hospitalization and mortality [82]. However, it is noteworthy that physical activity is not currently listed as a preventive measure for COPD in the Global Initiative for COPD [83].

### 5.2. Obesity

In 1997, obesity was recognized as a global epidemic due to its unprecedented incidence and considerable health and socioeconomic consequences. This condition affects individuals across all age groups and has steadily increased in developed and developing nations. Current estimates indicate that over 2.8 million individuals die annually because of complications associated with abnormal body weight across all age groups and this trend is steadily increasing in both the developed and developing world. Ectopic fat deposition (EFD) refers to the excessive accumulation of lipids in internal organs disrupting lung homeostasis through mechanical injury, alveolar structural damage, and inflammation. In obese individuals, adipose tissue accumulates in the thoracic visceral cavity, chest wall, and diaphragm, reducing the functional residual capacity (FRC) and expiratory reserve volume [84]. Fat accumulation in the airways disrupts the ultrastructure of the alveoli, contributing to tissue damage. Additionally dysanapsis, the mismatch between lung volume and the airways, is more common in obese children and is associated with worse asthma outcomes and poor treatment response [85]. Different studies have demonstrated the association between obesity and both, the development of asthma and its exacerbation in children [86]. Additionally, children with obesity and influenza infection have worse outcomes [87].

Adipose-derived adipokines, such as leptin and adiponectin, play a central role in lung inflammation. Leptin, secreted by adipose tissue and gastric mucosa in response to energy store availability, binds to the leptin receptors in macrophages, promoting phagocytosis and the production of pro-inflammatory and pro-fibrotic cytokines such as IL-1, Il-18, and TGF-β [88]. Obesity-related adiponectin deficiency contributes to mitochondrial dysfunction and a low-grade inflammatory state which has been implicated in COPD, asthma, bacterial and viral pneumonia, and aspergillosis [89]. A Korean study found that high adiponectin-to-leptin ratio (ALR) is associated with a rapid decline in lung function and an increased risk of airflow obstruction, suggesting its potential as a biomarker for predicting COPD [90]. The relationship between COPD prognosis and obesity is complex. A recent meta-analysis suggests that overweight individuals appear to have a better prognosis; however, this protective effect diminishes for individuals with a BMI exceeding 31 kg/m^2^ [91]. This phenomenon, known as the obesity paradox, should be interpreted with caution, as Body Mass Index (BMI) does not account for body composition, and an overestimation of airway obstruction in patients with morbid obesity is possible [91,92].

In summary, obesity negatively impacts lung function through mechanical constraints, airway remodeling, and chronic inflammation, while the complex relationship between obesity and respiratory diseases, including asthma and COPD, highlights the need for careful interpretation of its effects on lung health.

## 6. Environmental Factors and Lung Metabolism

Environmental factors significantly influence lung health throughout an individual’s lifespan, affecting respiratory function and accelerating the process of lung aging [93]. In fact, it is well described that initial exposures during fetal development and infancy, such as maternal smoking and air pollution, can impede lung development and increase the risk of respiratory diseases in later years. Prolonged exposure to pollutants such as particulate matter, ozone, nitrogen dioxide, and sulfur dioxide disrupts lung defenses and contributes to oxidative stress, inflammation, and structural damage [94]. Mechanisms such as DNA damage, mitochondrial dysfunction, and telomere shortening further accelerate lung aging and compromise respiratory health [93]. These environmental exposures often interact with genetic predispositions, exacerbating susceptibility to chronic respiratory diseases [95].

### 6.1. Particulate Matter

The term particulate matter (PM) encompasses many different small air particles that are a significant component of air pollution [96]. PM is defined as a particle with a diameter of less than 10 μm, capable of penetrating through the airways and are further subdivided into particles with a 2.5–10 μm diameter (PM coarse) and particles with a diameter less than 2.5 [97]. Exposure to PM can lead to significant metabolic changes in lung tissue due to oxidative stress, ferroptosis, inflammation, and cellular damage [98]. In addition, PM exposure can produce changes in lipid peroxidation, contributing to surfactant dysfunction and inflammation in alveolar cells [99]. In terms of inflammation, PM exposure has been related with an increase in TNF-α, IL-6, and IL-1β production, contributing to a chronic inflamed lung that may drive metabolic reprogramming in lung cells, similar to changes seen in cancer and fibrosis [100]. Finally, it has been shown that PM exposure has been linked to a shift towards aerobic glycolysis (Warburg effect), a hallmark of inflammation and some lung diseases (e.g., COPD, lung cancer).

### 6.2. Nitrogen Dioxide, Sulfur Dioxide, and Carbon Monoxide

Carbon monoxide (CO), sulfur dioxide (SO_2_), and nitrogen dioxide (NO_2_) are major air pollutants with significant effects on lung health [101]. Fossil fuel combustion and industrial activities are the primary sources of these pollutants, although natural and other human-related activities also contribute to their presence in the atmosphere [102]. The introduction of catalytic converters in vehicles has significantly reduced CO emissions. However, the increasing use of automobiles and household fossil fuel consumption continues to pose a public health risk [101]. CO exposure has been linked to impaired lung function, leading to reductions in FEV1 and forced vital capacity (FVC) regardless of exposure levels [103]. Furthermore, prior CO poisoning has been associated with an increased risk of COPD, lung cancer, and tuberculosis infection [104]. Similarly, both short- and long-term exposure to SO_2_ is associated with reduced lung function and is considered a risk factor for interstitial lung disease (ILD) and COPD [105,106]. NO_2_, a significant component of traffic-related air pollution, has also been linked to adverse respiratory outcomes. A meta-analysis found that a 10-μg/m^3^ increase in NO_2_ exposure is associated with a 4% increased risk of lung cancer [107]. Additionally, long-term NO_2_ exposure has been associated with a 2.0% increase in the relative risk (RR) of developing COPD [108].

### 6.3. Ozone

Ozone is a significant air pollutant that constitutes a risk factor for lung health. Ozone is formed in the troposphere by the interaction between sunlight and precursor gasses, mainly nitrogen oxides and volatile organic compounds (VOC) [109]. Ozone exposure has been shown to affect glutathione metabolism, reducing antioxidant defenses [110]. Moreover, increased protein oxidation and proteolysis contribute to chronic inflammation, lung injury, tissue remodeling, and fibrosis [111]. Epidemiological studies have consistently shown that elevated ambient ozone levels correlate with persistent respiratory symptoms and a decline in long-term lung function [112]. For example, it has been established that ozone aggravates COPD, leading to increased hospital admissions and mortality rates, while the data regarding its effect on asthma development remain inconclusive [113].

### 6.4. Lead Exposure

The WHO has categorized lead among the 10 most toxic chemicals, responsible for 1.5 million deaths in 2021 [114]. Lead (Pb) is absorbed via the gastrointestinal tract, the respiratory tract, and the skin. Notably, 40% of the inhaled lead remains in the alveoli [115]. Pb exposure can disrupt the balance between reactive oxygen species (ROS), which can lead to lipid peroxidation, protein oxidation, and DNA damage in lung cells, impairing cellular respiration and metabolic processes [116]. It produces inflammation and interferes with calcium signaling, as Pb ions can mimic Ca ones, interfering with calcium-dependent processes in cells and impacting cell signaling, enzyme regulation, and muscle contraction, which are critical for airway function [117]. Lead-induced metabolic disruptions have been associated with both acute and long-term conditions. For example, animal studies have indicated an association between Pb exposure and asthma severity; high levels of Pb resulted in increased levels of IgE and histamine, alongside a decreased level of interferon γ, suggesting an abnormal inflammatory response [118]. In the long term, Pb exposure has been associated with COPD, asthma exacerbation, and potentially contributes to lung cancer development [119,120].

### 6.5. Residential Radon

Radon 222 (^222^Rn), refers to radon gas that accumulates in homes and buildings, posing a significant indoor air quality and health risk. Radon is a radioactive gas formed from the natural decay of uranium in soil, rock, and water. It is colorless, odorless, and tasteless, making it difficult to detect without specialized testing. The primary sources of exposure are soil gasses and well-water [121]. Early epidemiological studies in uranium and silver miners established the relationship between lung malignancy and radon exposure [122]. In fact, it is a recognized carcinogen, and is the second leading risk factor for lung cancer after tobacco [123]. Apart from the numerous studies showing the relation between Rn exposure and lung cancer, Rn induces significant metabolic changes in the lung through oxidative stress, inflammation, and mitochondrial dysfunction, contributing to lung damage in patients with COPD [124,125].

All the above show that urgent air quality standards may be inadequate to effectively protect vulnerable populations, highlighting the need for more stringent regulations and public health interventions. The purpose of this review is not to exhaustively detail all environmental factors that can impact lung health, as the list is extensive and may be the subject of another review (see Figure 3). However, it is worth mentioning that microplastics are gaining increasing attention for their potential effects on human health and their accumulation in human tissues, particularly in the brain [126]. Regarding lung health, although the majority of inhaled microplastics are cleared via mucociliary clearance, retained particles can release toxic compounds such as vinyl chloride monomers, triggering inflammation, oxidative stress, and impaired lung function [127]. Future research should focus on elucidating the mechanisms by which microplastic exposure influences pulmonary metabolism and contributes to the development of lung diseases.

## 7. Conclusions Remarks

Metabolic flexibility, the ability of lung cells to adapt fuel usage to meet energy demands—is crucial for maintaining pulmonary homeostasis. The lungs’ high energy needs (for ciliary motion, surfactant production, immune defense, etc.) are normally met by shifting between substrates like glucose, fatty acids, and amino acids. This balance is readily disrupted by poor nutrition, sedentary lifestyles, obesity, and pollutant exposure. Inadequate intake of key micronutrients (e.g., vitamins A, D, C) weakens the airway epithelium and immune defenses, increasing susceptibility to infection and injury. Conversely, balanced diets rich in anti-inflammatory components—such as optimal omega-6/omega-3 fatty acid ratios—exhibit anti-inflammatory effects, protecting against oxidative damage and improving outcomes in conditions like COPD and asthma

Practicing physicians should therefore integrate nutritional assessment into respiratory care, correcting micronutrient deficiencies and promoting diets that support metabolic health, while using high-dose supplements cautiously (for example, excessive vitamin A proved risky in smokers). Regular physical activity is another cornerstone of lung health, as physical inactivity and obesity promote chronic inflammation and even mechanical breathing constraints. Paradoxically, in COPD patients, a higher body mass index is associated with improved survival—the “obesity paradox”—indicating that weight management must be individualized rather than one-size-fits-all. Clinicians should balance the benefits of weight reduction (improved ventilatory mechanics and metabolic profile) with this paradoxical protective effect of adipose tissue in severe disease.

Emerging evidence also suggests that *when* we eat matters; meal timing and circadian-aligned feeding (chrono-nutrition) can influence lung metabolism and inflammation. Interventions like time-restricted feeding or fasting may thus become novel adjuncts to support lung function, though more research is needed before clinical recommendations can be made.

Environmental exposures are equally critical. Although regulatory interventions have helped reduce pollutant exposure, further efforts are needed to adequately protect the general population and individuals with pre-existing conditions who remain at significant risk. Further research into how these pollutants affect lung homeostasis at a molecular level could help regulatory bodies implement targeted preventive strategies, develop specific biomarkers of population and higher risks, and ultimately reduce exposure-related health hazards. We believe that stronger action from governments and health agencies is urgently needed to enforce stricter air quality standards, implement targeted public health policies, and ensure effective protection for those most susceptible to respiratory harm.

Future research should prioritize a deeper understanding of the metabolic pathways most affected in respiratory conditions. By integrating metabolic studies with clinical research, we can identify biomarkers and therapeutic targets to guide personalized interventions. Ultimately, a multidisciplinary approach combining nutrition, physical activity, and environmental monitoring could offer comprehensive strategies for preserving lung function and mitigating disease progression. 

## Figures and Tables

**Figure 1 nutrients-17-00954-f001:**
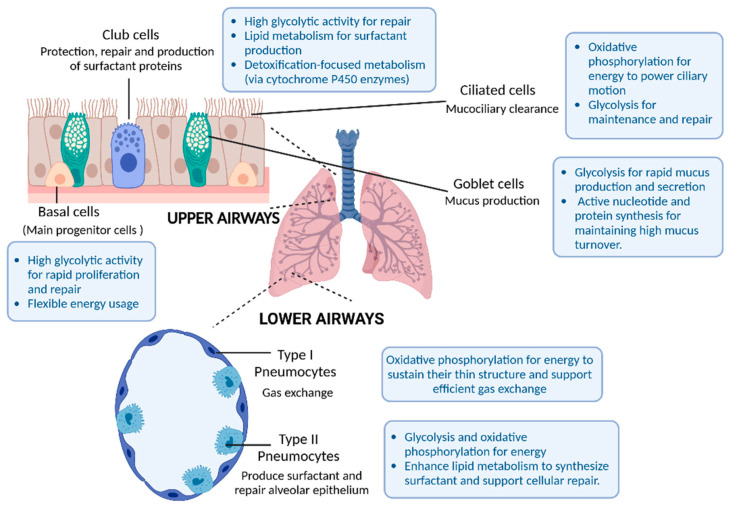
Main epithelial cell types in the lungs, highlighting main functions and metabolic pathways.

**Figure 2 nutrients-17-00954-f002:**
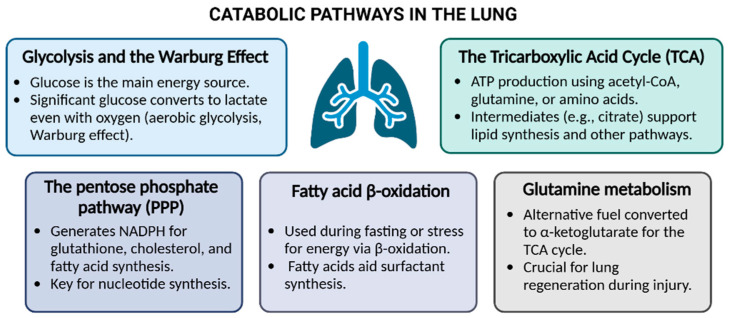
Main catabolic pathways involved in lung metabolism.

**Figure 3 nutrients-17-00954-f003:**
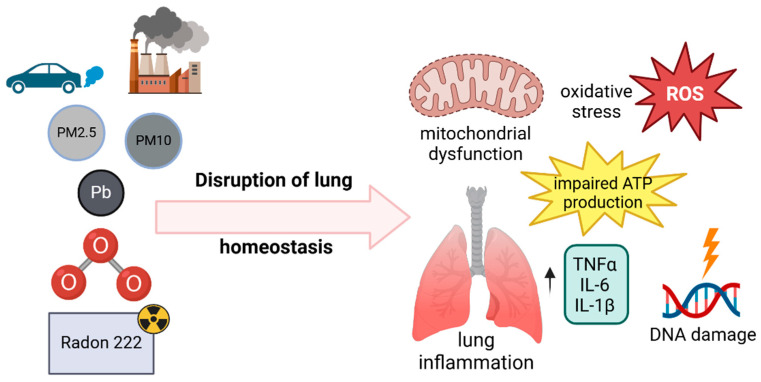
Mechanism of lung homeostasis disruption by environmental pollutants. Environmental pollutants, including particulate matter (PM), nitrogen dioxide (NO_2_), sulfur dioxide (SO_2_), carbon monoxide (CO), ozone (O_3_), lead (Pb), and radon (Rn), disrupt lung homeostasis through four key mechanisms: mitochondrial dysfunction (oxidative stress impairs ATP production, causing metabolic shifts and energy deficits, oxidative stress); reactive oxygen species (ROS) damage lipids, proteins, and DNA, inflammation; pollutants activate pro-inflammatory cytokines (TNF-α, IL-6, IL-1β), triggering immune responses and tissue damage; and DNA damage (radon induces DNA mutations), increasing lung cancer risk. Chronic oxidative stress and inflammation further promote malignancy. These mechanisms interact, accelerating lung disease progression and increasing susceptibility to COPD, asthma, pulmonary fibrosis, and lung cancer.

## Data Availability

Not applicable.

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
