# Peer review of "Nutrition, Lifestyle, and Environmental Factors in Lung Homeostasis and Respiratory Health"

_nutrients, 2025, doi:10.3390/nu17060954_

Round 1
Reviewer 1 Report
Comments and Suggestions for Authors
The manuscript entitled “Nutrition, Lifestyle, and Environmental Factors in Lung Homeostasis and Respiratory Health” presents a comprehensive study on the influence of various factors on metabolism and lung function. In light of the recent coronavirus pandemic and its devastating impact on the respiratory system, the thematic scope taken up by the authors seems to be important and interesting. This is confirmed by the selection of factors affecting the lungs included in the manuscript, i.e., modifiable factors in a significant part.
However, I would call the selection of some of the references a shortcoming. The “References” section includes 165 items, which is a very large number. Unfortunately, almost 30 percent (46 items) are relatively “old” items, i.e. from before 2015. Additionally, 13 items are from before 2000. Considering the extensiveness of this section, I would suggest removing these “old” references and leaving the more current literature. I also have concerns about the authors of the manuscript regarding lifestyle factors that affect the respiratory system. More specifically, it seems incorrect to me to single out nutrition (section 4: “How nutrition affects lung metabolism”) as a non-lifestyle component. Lifestyle is included in a separate section: “5. Lifestyle and lung metabolism.” I suspect that the authors of the manuscript intended to single out nutrition as a separate and therefore very important factor influencing respiratory function. However, I would suggest clearly indicating that it is a lifestyle factor. This would of course require modification of the title of the manuscript, the content of the abstract, the stated purpose, as well as the titles of some sections.
Below are the remaining comments and remarks that I consider important to include in the development of the revised version of the manuscript. I present them in the order in which they appear in the manuscript, not in terms of their importance.
- Section 1 “Introduction”:
I suggest making corrections in the last two paragraphs due to the authors' use of the future tense instead of the past tense (lines 51-55: “We will begin with a brief overview of the lung’s structure and anatomy to contextualize the discussion on cellular energy consumption and metabolic pathways. Additionally, we will analyze how dietary habits, lifestyle choices, and environmental factors affect lung metabolism, emphasizing the potential role of nutritional supplements in preventing or reversing metabolic dysfunction” and lines 61-63: “Finally, we will review emerging evidence that links alterations in cellular metabolism to the pathobiology of common respiratory diseases, including chronic obstructive pulmonary disease (COPD), asthma, and pulmonary fibrosis.”). The manuscript has already been written, the authors have already reviewed the references, etc., so using the future tense (“will”) is incorrect.
The same applies to the sentence in lines 210-212: “In the following sections we will explore in detail the potential role of nutritional supplements in both healthy individuals and respiratory patients.”
- The sentence in lines 97-99: “The goal of this review is not to provide details of the various characteristics of different lung cell populations. Instead we have briefly summarised its main functions and the specific metabolic patterns of each epithelial cell type (see Figure 1)” I consider an unnecessary explanation.
- The way of inserting references should be corrected throughout the text: in the text of the manuscript, reference numbers should be placed in square brackets [ ]. In addition, two or more references to references should be enclosed in one common bracket – this is not the case in line 451: “(106) (36)”. It is also possible that “(36)” is not a reference to reference 36 in the References section?
- Figure 1: the fragment enclosed in brackets in the caption of this figure is probably unnecessary? Line 113: "(text within the blue square)".
- All abbreviations used should be explained in the first place they appear in the text. Meanwhile, the authors either did not explain some abbreviations at all (e.g. NAD+, NADH, TCA, LDH, NADPH, MAPK, Wnt, FEV1, FVC and others) or inserted these explanations when the abbreviation was used again (e.g. the term "intermittent fasting" was presented as the abbreviation "IF" only on the second use - lines 367 and 377 and the term "body mass index" was presented as an abbreviation only on the third use - lines 413, 457, 458 and others) or inserted duplicate explanations (e.g. "PPP" in line 145 explained earlier on line 139 and "COPD" in lines 286 and 311 explained earlier on line 63 and others). This requires correction by the authors of the manuscript. Additionally, the use of the subscript in the abbreviation FEV1 requires standardization - this abbreviation appears in the text both without the subscript and with the subscript.
- In line 125 and in Figure 1, the authors used the term "Glycosis" - is this the correct name? Shouldn't it be "Glycolysis"? This would be indicated by the text of the manuscript.
- Line 303: in the underlined text fragment, replace "acid" with "acids". Line 304: replace "fats" with "fatty acids".
- Sentences from lines 303-305: “Omega-3 fatty acids, primarily consisting of eicosapentaenoic acid (EPA) and docosahexaenoic acid (DHA), are polyunsaturated fats that have well-documented anti-inflammatory and antioxidant properties” and from lines 307-308: “Omega-3 fatty acids play a crucial role in the regulation of lung inflammation, oxidative stress, and immune function (61)” contain repetitions of content. This should be corrected.
- Subsection 4.2 Practical Dietary Recommendations for Lung Health
Sentence in lines 330-331: “Certain foods can reduce inflammation, enhance lung capacity, and improve overall breathing efficiency” – instead of the term “certain foods” I suggest using “certain nutrients” because the action indicated by the authors concerns certain nutrients, not whole food products.
In this subsection, which is a summary of section 4 (How nutrition affects lung metabolism), the authors listed ingredients that they had not mentioned at all before, i.e.: flavonoids (line 333), high-fiber foods (line 334), trans fats (line 338), magnesium (line 340). Additionally, the term "trans fats" is incorrect - instead it should be "trans fatty acids".
In lines 349-350, the authors used the term "healthy fats" - what exactly does it mean? This is a term used in everyday language (similarly to e.g. healthy lifestyle or healthy diet), but it should not be used in scientific texts.
- The footnote to Table 1 should include an explanation of the abbreviation "COPD".
- The subsection 5.2. Obesity highlighted in section 5. Lifestyle and lung metabolism is incorrect because obesity is not an element of lifestyle. It is a consequence of, among other things, leading a certain lifestyle.
- Line 519: "(VOCs.)" - the dot should be removed.
- Line 534: "Lead Pb" - "PB" should be enclosed in brackets.
- Line 539: "Ca2+" - I consider it unnecessary to present it as an ion, it is enough to use the name of the element.
- Figure 3:
due to the fact that this figure summarizes the content of section 6. Environmental factors and lung metabolism, the reference to it should be placed in this section. However, the authors incorrectly inserted this reference only in section 7. Conclusion remarks.
This figure also contains an error: instead of "inflamation" it should be "inflammation".
The mechanisms listed in the caption to this figure: 1. Mitochondrial Dysfunction, 2. Oxidative Stress, 3. Inflammation, 4. Carcinogenesis should also be added to the figure, since this figure is supposed to illustrate the mechanisms of the negative impact of environmental pollutants on the lungs.
- Section 7. Conclusion remarks
The sentence in lines 586-588: “The lungs, beyond their role in gas exchange, depend on a wide variety of specialized cells, including epithelial cells, immune cells like macrophages, and structural cells such as fibroblasts, each with distinct metabolic requirements” requires modification – I consider it unnecessary to insert the fragment “beyond their role in gas exchange”.
This section, as a summary section, should no longer contain references to references – the reference “(41)” from line 603 should be removed.
This section is too long. It should contain a concise summary of the presented results. In addition, the authors placed great emphasis on supplementation in this part of the text. Firstly, the manuscript text did not indicate this, and secondly, it is rather worth emphasizing the role (or influence) of proper diet balancing.
- The entire References section requires standardization and adjustment to the journal's requirements. In particular: the number of included authors of individual references should be corrected, dots should be inserted after the initials of the authors' names, unnecessary capital letters should be removed from the titles of manuscripts, the year of publication should be bold, the volume number should be entered in italics, Abbreviated Journal Names should be inserted in italics.
I do not consider myself competent to assess the quality of English language, but the review form required me to select appropriate options.
Author Response
Thanks for your valuable comments regarding the manuscript. We really appreciate them.
Regarding to your first comment: However, I would call the selection of some of the references a shortcoming. The “References” section includes 165 items, which is a very large number. Unfortunately, almost 30 percent (46 items) are relatively “old” items, i.e. from before 2015. Additionally, 13 items are from before 2000. Considering the extensiveness of this section, I would suggest removing these “old” references and leaving the more current literature.
We acknowledge that there was an excessive number of references, and we have considerably reduced them. We are aware that there are still many "old" references, especially in the first part of the article, where we discuss basic metabolic pathways, as we wanted to mention the first time they were discovered. In any case, we thank the reviewer for their suggestion, and we have reduced the total number of references, particularly the older ones.
I also have concerns about the authors of the manuscript regarding lifestyle factors that affect the respiratory system. More specifically, it seems incorrect to me to single out nutrition (section 4: “How nutrition affects lung metabolism”) as a non-lifestyle component. Lifestyle is included in a separate section: “5. Lifestyle and lung metabolism.” I suspect that the authors of the manuscript intended to single out nutrition as a separate and therefore very important factor influencing respiratory function. However, I would suggest clearly indicating that it is a lifestyle factor. This would of course require modification of the title of the manuscript, the content of the abstract, the stated purpose, as well as the titles of some sections.
We agree with the reviewer and have now changed the title of section 5 to "Other Lifestyle Factors Affecting Lung Metabolism." This way, we first discuss nutrition in section 4, and then in section 5, we cover other lifestyle factors apart from nutrition. We hope this makes it clearer.
Other comments:
- Section 1 “Introduction”:
I suggest making corrections in the last two paragraphs due to the authors' use of the future tense instead of the past tense (lines 51-55: “We will begin with a brief overview of the lung’s structure and anatomy to contextualize the discussion on cellular energy consumption and metabolic pathways. Additionally, we will analyze how dietary habits, lifestyle choices, and environmental factors affect lung metabolism, emphasizing the potential role of nutritional supplements in preventing or reversing metabolic dysfunction” and lines 61-63: “Finally, we will review emerging evidence that links alterations in cellular metabolism to the pathobiology of common respiratory diseases, including chronic obstructive pulmonary disease (COPD), asthma, and pulmonary fibrosis.”). The manuscript has already been written, the authors have already reviewed the references, etc., so using the future tense (“will”) is incorrect.
The same applies to the sentence in lines 210-212: “In the following sections we will explore in detail the potential role of nutritional supplements in both healthy individuals and respiratory patients.”
We are grateful to the reviewer for this comment, but in this specific case, we hope they understand that we prefer to leave it as it is. This is part of my personal writing style—I prefer to use the future tense to capture the reader's attention, give them a sense of what is coming, and help them stay more engaged. However, if the editor also prefers that we change it to the past tense, we are happy to do so.
- The sentence in lines 97-99: “The goal of this review is not to provide details of the various characteristics of different lung cell populations. Instead we have briefly summarised its main functions and the specific metabolic patterns of each epithelial cell type (see Figure 1)” I consider an unnecessary explanation.
We have removed this sentence as suggested.
- The way of inserting references should be corrected throughout the text: in the text of the manuscript, reference numbers should be placed in square brackets [ ]. In addition, two or more references to references should be enclosed in one common bracket – this is not the case in line 451: “(106) (36)”. It is also possible that “(36)” is not a reference to reference 36 in the References section?
We apologise for this, and we have now made sure that all the references are corrected. Also, when having various references inserted together have been enclosed in one common bracket.
- Figure 1: the fragment enclosed in brackets in the caption of this figure is probably unnecessary? Line 113: "(text within the blue square)".
We have removed this sentence as suggested.
- All abbreviations used should be explained in the first place they appear in the text. Meanwhile, the authors either did not explain some abbreviations at all (e.g. NAD+, NADH, TCA, LDH, NADPH, MAPK, Wnt, FEV1, FVC and others) or inserted these explanations when the abbreviation was used again (e.g. the term "intermittent fasting" was presented as the abbreviation "IF" only on the second use - lines 367 and 377 and the term "body mass index" was presented as an abbreviation only on the third use - lines 413, 457, 458 and others) or inserted duplicate explanations (e.g. "PPP" in line 145 explained earlier on line 139 and "COPD" in lines 286 and 311 explained earlier on line 63 and others). This requires correction by the authors of the manuscript. Additionally, the use of the subscript in the abbreviation FEV1 requires standardization - this abbreviation appears in the text both without the subscript and with the subscript
We apologise for this, and we have made sure that all the abbreviations are explained the first time that they appear in the text. Also, we have standarised the abbreviation FEV1.
- In line 125 and in Figure 1, the authors used the term "Glycosis" - is this the correct name? Shouldn't it be "Glycolysis"? This would be indicated by the text of the manuscript.
We apologise for this typo, as the correct term is glycolysis, so we have changed accordingly.
- Line 303: in the underlined text fragment, replace "acid" with "acids". Line 304: replace "fats" with "fatty acids".
This has been replaced as suggested.
- Sentences from lines 303-305: “Omega-3 fatty acids, primarily consisting of eicosapentaenoic acid (EPA) and docosahexaenoic acid (DHA), are polyunsaturated fats that have well-documented anti-inflammatory and antioxidant properties” and from lines 307-308: “Omega-3 fatty acids play a crucial role in the regulation of lung inflammation, oxidative stress, and immune function (61)” contain repetitions of content. This should be corrected.
We thank the reviewer for highlighting this. However, we don’t believe is a repetition of content we are describing the effects in general the first time, and talking specifically about the lung, in the second part.
- Subsection 4.2 Practical Dietary Recommendations for Lung Health
Sentence in lines 330-331: “Certain foods can reduce inflammation, enhance lung capacity, and improve overall breathing efficiency” – instead of the term “certain foods” I suggest using “certain nutrients” because the action indicated by the authors concerns certain nutrients, not whole food products.
We have kindly accepted this suggestion.
In this subsection, which is a summary of section 4 (How nutrition affects lung metabolism), the authors listed ingredients that they had not mentioned at all before, i.e.: flavonoids (line 333), high-fiber foods (line 334), trans fats (line 338), magnesium (line 340). Additionally, the term "trans fats" is incorrect - instead it should be "trans fatty acids".
We have changed the term trans fatty acids. Related to the first suggestion, we don´t quite agree that the subsection is a summary of section 4. In section 4, we have described the effects on metabolism of some nutrients. Understandably, we can not describe them all. Then, in the next subsection, we have done some suggestions, including nutrients that may have not been described in section 4.
In lines 349-350, the authors used the term "healthy fats" - what exactly does it mean? This is a term used in everyday language (similarly to e.g. healthy lifestyle or healthy diet), but it should not be used in scientific texts.
We agree with the reviewer, and we have changed the term for unsaturated fats and essential fatty acids.
- The footnote to Table 1 should include an explanation of the abbreviation "COPD".
This has been added.
- The subsection 5.2. Obesity highlighted in section 5. Lifestyle and lung metabolism is incorrect because obesity is not an element of lifestyle. It is a consequence of, among other things, leading a certain lifestyle.
We acknowledge the intention of the reviewer, and we appreciate the comment. However, I do not fully agree, as I think that obesity is an element of lifestyle, though it is also influenced by genetic, metabolic, and environmental factors. Lifestyle choices—such as diet, physical activity, sleep patterns, and stress management—play a significant role in the development and progression of obesity. We have not received this comment from the editor, so we have decided to leave it as it is unless explicitly suggested by the editor.
- Line 519: "(VOCs.)" - the dot should be removed.
The dot has been removed.
- Line 534: "Lead Pb" - "PB" should be enclosed in brackets.
This has been done.
- Line 539: "Ca2+" - I consider it unnecessary to present it as an ion, it is enough to use the name of the element.
Thanks for this comment. We indeed doubted about the best way to name Calcium. We have changed as suggested by the reviewer.
- Figure 3: due to the fact that this figure summarizes the content of section 6. Environmental factors and lung metabolism, the reference to it should be placed in this section. However, the authors incorrectly inserted this reference only in section 7
We have also added the reference to figure 3 in section 6.
This figure also contains an error: instead of "inflamation" it should be "inflammation".
This has been changed.
The mechanisms listed in the caption to this figure: 1. Mitochondrial Dysfunction, 2. Oxidative Stress, 3. Inflammation, 4. Carcinogenesis should also be added to the figure, since this figure is supposed to illustrate the mechanisms of the negative impact of environmental pollutants on the lungs.
Thanks for this suggestion. We have adjusted the figure legend.
- Section 7. Conclusion remarks
The sentence in lines 586-588: “The lungs, beyond their role in gas exchange, depend on a wide variety of specialized cells, including epithelial cells, immune cells like macrophages, and structural cells such as fibroblasts, each with distinct metabolic requirements” requires modification – I consider it unnecessary to insert the fragment “beyond their role in gas exchange”.
I personally think this is a matter of style, and we are not very sure why this sentence should be removed. At any case, we have rewritten the conclusion remarks and this sentence doesn´t appear any more.
This section, as a summary section, should no longer contain references to references – the reference “(41)” from line 603 should be removed.
Thanks for noticing this. The reference has been removed.
This section is too long. It should contain a concise summary of the presented results. In addition, the authors placed great emphasis on supplementation in this part of the text. Firstly, the manuscript text did not indicate this, and secondly, it is rather worth emphasizing the role (or influence) of proper diet balancing.
We agree with this comment, and we have rewritten the conclusion section accordingly to both the reviewers and editor´s comments.
- The entire References section requires standardization and adjustment to the journal's requirements. In particular: the number of included authors of individual references should be corrected, dots should be inserted after the initials of the authors' names, unnecessary capital letters should be removed from the titles of manuscripts, the year of publication should be bold, the volume number should be entered in italics, Abbreviated Journal Names should be inserted in italics.
We are sorry for this. We have used a reference system (Zotero), following Vancouver. We hope that the journal's formatting and layout team can handle this aspect and take it from here.
Reviewer 2 Report
Comments and Suggestions for Authors
The authors tackle a very complicate and up to date problem for the interactions between external and internal body environments and the condition and function of respiratory system. The topic is very original and needs a lot of research in the light of the functions of microbiome and nutrition and their regulatory effects on the human body.
The manuscript provides a well-structured and comprehensive review of the current knowledge on the topic and gives a broad view on it which is difficult to find in other articles. I hope that future articles will go into detail about the different "facets", set in the multidisciplinary and holistic approach, proposed by the authors.
I do not have any recommendations about the methodology of this manuscript. Of course it is impossible to include all the factors, involved in the interactions of concern, but it seems to me that for example the impact of low molecule weight substances like chlorine, disinfectants, and food preservatives should could also be considered as modulating factor.
The conclusions are consistent with the evidence and arguments provided. The reverences are concise and appropriate. I do not have remarks on the tables and figures.
Author Response
Thank you to the reviewer for the comments that are very appreciated.